# Variables Influencing Professors’ Adaptation to Digital Learning Environments during the COVID-19 Pandemic

**DOI:** 10.3390/ijerph19063732

**Published:** 2022-03-21

**Authors:** Diego Vergara-Rodríguez, Álvaro Antón-Sancho, Pablo Fernández-Arias

**Affiliations:** 1Department of Mechanical Engineering, Catholic University of Ávila, C/Canteros, s/n, 05005 Avila, Spain; pablo.fernandezarias@ucavila.es; 2Department of Mathematics and Experimental Science, Catholic University of Ávila, C/Canteros, s/n, 05005 Avila, Spain; alvaro.anton@ucavila.es

**Keywords:** stress, digital competence, digitization process, COVID-19 pandemic, psychological effects

## Abstract

This paper conducts quantitative research on the influence of the COVID-19 pandemic on the adaptation to digital learning environments (DLE) of a group of 908 university professors. We compared the perceptions of participants who were digital natives (born before 1980) with those of digital immigrants (born after 1980). For this purpose, a validated survey was used and the obtained responses statistically analyzed. The results show a negative correlation between pandemic stress and the digital competence of professors and their adaptation skills to digital environments, which although weak for both of the two groups compared are stronger for digital immigrants. Both self-confidence and digital competence show a positive influence on the perception of adaptation of skills to DLE, and this is weaker in digital natives. Gaps were identified by gender and area of knowledge of the participants; consequently, the need to carry out training actions for university faculty on skills linked to their digital competence in teaching is clear.

## 1. Introduction

On 30 January 2020, the World Health Organization (WHO) declared the COVID-19 outbreak an international public health emergency, and on 11 March 2020, a pandemic [1]. The confinement suffered in many countries during the year 2020 as a consequence of the pandemic generated by the SARS-CoV-2 virus had an immediate impact on human behavior [2] and brought about rapid changes in the activity of different professional profiles [3], including teachers carrying out their teaching activity at different educational levels. The impossibility of continuing to teach on-site at different educational levels generated a 180° transformation of the educational system in a matter of hours [4]. Over 1.5 billion students from around the world were affected by school or university closures during the early phase of the COVID-19 pandemic in 2020, and distance learning was introduced for many pupils [5]. About 94% of the world’s student population and millions of teachers around the world adapted to on-line teaching almost immediately in order to be able to continue teaching [6,7], including in laboratory practice [8,9].

This transformation brought to light the weaknesses of the educational system at the time, especially lack of virtual learning resources, as the system was based mainly on the presence of all the actors involved in the teaching–learning process [10]. This transformation even entailed the temporary closure of several schools and universities [11,12]. In order to carry out this rapid adaptation of the educational system towards virtuality, it was necessary to invest considerable economic and technical resources and to rapidly acquire numerous electronic and peripheral devices, i.e., web cams, headsets, and virtual communication platforms. The high development of Information and Communication Technologies (ICT) during the years prior to the pandemic was key to coping with the confinement situation [13].

However, it was not enough to acquire numerous ICT; teachers had to face the stress of teaching in a scenario they had never experienced before [14,15]. Many of them at many different educational levels were unable to cope with the situation of increased stress and isolation [16], even abandoning their professional careers [17]. Other teachers coped with the situation by applying active and effective learning strategies in order to reduce the risk of increasing their stress and burnout [18,19,20]. These quick pedagogical decisions taken by teachers in the first moments of the confinement were very important in the subsequent distance teaching–learning process [21].

Therefore, in order to overcome this situation many teachers have transformed their teaching methods, relying on various competencies and skills (Figure 1): (i) methodological competence [19]; (ii) soft skills [22]; and (iii) digital competence [23].

Methodological competence, as before confinement, remained important. Teachers had to have numerous resources and methodologies to transmit timely knowledge to their students, regardless of the medium or environment in which they did it [24]. Similarly, soft skills play a fundamental role in this process of digital teacher adaptation. Soft skills are a series of interpersonal attributes and personal qualities that are linked to personality traits, including abilities such as flexibility, self-control, communication, empathy, and emotional intelligence [22,25]. These competencies, very in demand in the 21st century, have been fundamental throughout the confinement for three reasons: (i) to help students cope with the situation and their professional future, as well as to understand their personal circumstances [26,27]; (ii) coping with possible stress and frustration generated in the teachers themselves [28] and in their students throughout the confinement [29]; and (iii) meeting the challenges of e-learning education [30].

In terms of digital skills, teachers sometimes had to apply skills that in most cases they had not had to apply before, such as data processing and protection; communication in digital environments, sharing online resources, connecting and collaborating with other people, reworking content and evaluation tests, developing virtual environments, making artistic productions or multimedia content, computer programming, etc. [31,32].

In the same way, they had to face the abrupt virtualization of the educational system, sometimes with a shortage of resources, both ICT and correct workplaces. This situation was seen by the teachers themselves as an assessment of their professionalism [33].

Different studies over the last few months have developed the fundamental role played by soft skills in teachers’ and students’ ability to face the situation [26]. Similarly, different publications have analyzed the levels of stress and burnout in teachers of different educational levels in the pandemic scenario [1,34,35], as well as the psychological problems such as depression, anxiety, isolation, and loneliness that the pandemic has caused [36,37,38], detecting differences in the levels of stress generated in teachers according to gender [39], place of residence [40,41], and cultural and social characteristics [42]. Other studies have analyzed in teachers the relationship between work hours, work–family balance and quality of life (QoL) [43,44], although teachers already reported a low perception of QoL before the COVID-19 pandemic with a significant impact on mental and physical health due to various stress factors associated with work overload [45].

On the other hand, other studies analyzed teachers’ perception of their own self-efficacy [46] and their level of digital competence [31,47,48], identifying the need to implement professional development that allows the development of an optimal level of digital competence in the teaching community [49], as well as the effects that the use of ICT has on teachers [50] and the future challenges that its implementation implies [51]. On the other hand, having a low level of digital competence means an increase in perceived stress as well as a reduction in teachers’ motivation [33,52].

Other articles have analyzed the educational development of students throughout the pandemic [53,54], their opinion on the teaching received during COVID-19 confinement [55,56], their perception of the challenges and opportunities that this pandemic has offered to online training [57], and the changes in their habits and routines as well as in the way they relate to each other [27]. According to Cicha et al. [58], both the feeling of pleasure and the sense of self-efficacy are the most influential factors in convincing students to switch from face-to-face to virtual learning models.

For other authors, once the pandemic is over numerous opportunities will present themselves for the educational system, which will continue to expand online learning. Similarly, educational institutions will organize themselves more systematically to pursue the aspects of technology-based learning that have been found to be the most useful [59,60]. Mok et al. [61], on the other hand, identify that this pandemic and the implementation of the on-line education system will intensify social and economic inequalities in the various higher education systems.

Despite the large number of studies related to teacher stress, no research has been found that relates the different levels of development and application of soft skills and digital competence with the level of stress generated in teachers throughout the pandemic caused by COVID-19. Nor are there any studies that address this same issue by comparing the pandemic stress suffered by university teachers, differentiating between immigrants and digital natives with the perception of their adaptation to the needs of digitization of learning environments caused by the pandemic.

The concepts of digital native and digital immigrant have their origin in the work of Prensky [62,63] (Figure 2), who defines digital natives as those individuals who have grown up in a digitized world and have been surrounded by computers, telephones, video games, and other tools that have constituted their surrounding environment from birth. In Prensky’s terms [62], the digital world is for them like their native language. In contrast, digital immigrants are those who have had to incorporate digital tools at another point in their lives in their daily or professional activities, and for whom digitalization is not their native language but a second language that they have had to learn. These two digital generations are distinguished by age and in relation to their level of immersion in a fully digitized society. Prensky [62] dates the time point that differentiates the two digital generations to 1980; those born after that date are considered digital natives and the rest are digital immigrants. Following Prensky’s original work, many studies interested in analyzing the digital divide between the aforementioned generations have taken 1980 as the date that differentiates them [64,65,66,67,68,69,70,71,72,73,74]. The distinction between digital generations depends on age as well as on the socio-cultural aspects involved. For example, a young person living in a low-income context could be considered a digital immigrant because of the limited access to the use of digital technologies that, foreseeably, he or she may have had due to economic difficulties. However, for the sake of clarity the year of birth will be taken in this work as the main criterion to distinguish digital immigrants from digital natives, given that the target population of the study, namely university professors, is part of a homogeneous socio-cultural sector.

Thus, the main aim of this paper is to study the level of self-perception of digital competences and soft skills related to self-confidence and the ability to adapt to digital learning environments throughout the process of teaching virtualization during the COVID-19 pandemic in a population of university professors. The influence of the stress caused by the pandemic on university professors is analyzed along with the aforementioned self-perceptions. In addition, the paper analyzes the differences that exist between professors of the two digital generations (digital natives and digital immigrants) with respect to the previously-mentioned aspects and discusses the degree to which this generational variable is discriminative for stress caused by the pandemic in professors and their level of adaptation to digital environments.

## 2. Materials and Methods

This section explains the main features of the methodological design of the research. Specifically, the following issues are addressed: (i) description of the participants in the study; (ii) statement of the objectives and variables of the work; (iii) description of the research instrument; and (iv) explanation of the research procedure.

### 2.1. Participants

A total of 908 university professors participated in the study, chosen by means of a non-probabilistic convenience sampling process. The professors came from 21 different countries (Figure 3): Argentina, Bolivia, Brazil, Chile, Colombia, Costa Rica, Cuba, Dominican Republic, Ecuador, El Salvador, Guatemala, Honduras, Mexico, Nicaragua, Panama, Paraguay, Peru, Puerto Rico, Spain, Uruguay, and Venezuela. Among them were represented professors of both genders and those whose academic activity was developed in all areas of knowledge. The professors were contacted by e-mail and were sent the survey used as an instrument through GoogleForms^TM^. All participants answered the survey voluntarily, freely, and anonymously and their answers and all answers were checked for validity.

### 2.2. Objectives and Variables

The general objective of this research was to analyze the impact of the COVID-19 pandemic and the consequent increase in the use of digital learning environments (DLE) on the self-confidence of professors as well as the self-perception of the digital competence and adaptation skills of university professors in the use of these DLE and whether there were differences in this regard between digital native and digital immigrant professors. Specifically, the following objectives were pursued: (i) to study the impact of the pandemic on the self-confidence of professors, the digital competence, adaptation skills and professional aspects related to the use of DLE, and the level of pandemic stress in the professional development of participants; (ii) to analyze the correlations between the perceptions of self-confidence, digital competence, and professional issues related to the development of digital competence during the pandemic and the pandemic stress of professors on their adaptation skills to DLE; (iii) to identify significant differences between digital native and digital immigrant professors in terms of their self-confidence, self-concept of digital competence, and adaptation skills to DLE and the level of pandemic stress of the professors; and (iv) to analyze whether there were gender gaps or significant differences by area of knowledge in the self-perception of the impact of the pandemic within professors of both generations.

The main independent variable of the study was the digital generation (digital immigrant or digital native) of the participants. A digital native was defined as a participant born in 1980 or later, and a digital immigrant was defined as a participant born prior to that date. The independent variables of gender and area of knowledge were considered as well (Figure 4). The first variable referred to the sociological profile of the participants and the second to their academic specialization. Gender was a dichotomous variable, and its possible values were male or female. Area of knowledge was polytomous; the values considered for this variable were the following: Arts and Humanities, Sciences, Health Sciences, Social and Legal Sciences, Engineering, and Architecture.

The dependent variables studied in this research were the following: (i) assessment of the aspects indicated on digital competence; (ii) assessment of the aspects indicated on professional aspects of the use of DLE; (iii) self-confidence during the pandemic; (iv) skills of adaptation to DLE, according to the needs imposed by the pandemic; and (v) impact of pandemic stress on professors’ work.

### 2.3. Instrument

In this study, a survey of our own design was used to measure the self-concept of the participants with respect to the dependent variables considered. The survey consisted of 32 questions or items divided into five parts corresponding to the five variables described, as shown in Table 1. The items were rated by the participants on a Likert scale from 1 to 5, with 1 corresponding to the lowest valuation and 5 to the highest valuation.

### 2.4. Procedure

In this work, quantitative empirical research was carried out based on the results of the survey shown in Table 1. The following phases were followed: (i) research approach and instrument design; (ii) collection of the answers to the survey; and (iii) validation and analysis of the results. After contacting the participants, collecting the answers, and checking their validity, the instrument was validated from different points of view. First, an Exploratory Factor Analysis (EFA) was carried out to determine the factors of the instrument that explain the answers to the survey. The theoretical model defined by the EFA was confirmed by the indices of the Confirmatory Factor Analysis (CFA). The different factors defined were used to determine the different subscales of the survey. The psychometric validation of the instrument was carried out by means of Pearson correlation coefficients of the subscales defined both among themselves and with respect to the overall survey. Analysis of the convergent validity was carried out through the average variance extracted (AVE). Finally, the internal consistency of the instrument was verified with Cronbach’s alpha parameters and composite reliability (CR) of the subscales. A descriptive analysis of the answers was made based on the means and standard deviations of the different subscales. Pearson correlation coefficients were computed to analyze the degree of dependence between the different subscales of the survey. The *t*-test and Levene’s test were applied to identify gaps in the mean answers and standard deviations, respectively, when participants were differentiated by their digital generation (digital immigrants or digital natives). Likewise, the digital generation variable was crossed with the rest of the independent variables and the existence of gaps in the mean answers when differentiating by each independent variable within the participants of each digital generation, and identified by means of the Multifactor ANOVA test (MANOVA). All statistical tests were performed with a significance level of 0.05.

## 3. Results

This section presents the main results of the quantitative analysis of the survey answers. For the sake of clarity, the section has been divided into three parts: (i) results about the profile of the participants; (ii) analysis of the validation of the instrument; and (iii) answers from the survey.

### 3.1. Sample of Participants

The distribution of the participants, distinguishing between digital generation (immigrants or natives), is shown in Table 1. While digital immigrants (50.66%) slightly outnumber digital natives (49.34%), the goodness-of-fit test statistics permit the assumption that the distribution of participants differentiating by digital generation is approximately homogeneous (chi-square = 1.7445, *p*-value = 0.1866). In addition, the Pearson independence test statistics (Table 2) indicate that the distributions of the considered variables (gender and area of knowledge) are independent of the distribution of participants by digital generation.

From the data in Table 2, it can be deduced that while female professors are more common than males in both digital generations, this gap is more pronounced within natives. On the other hand, digital natives are more frequently professors of Humanities, while within digital immigrants there is a majority of professors of Social Sciences and Engineering.

### 3.2. Factor Analysis and Validation of the Instrument

An EFA was carried out with Varimax rotation on the answers to the survey, the factor weights of which led to the conclusion that none of the items proposed was superfluous and made it possible to identify five factors to explain the survey as a whole (chi-square = 4833.75, df = 625, *p*-value = 0.7625). Table 3 shows the highest factorial weights for each item, which are those that define each of the factors into which the survey is divided. Due to the specific contents of the questions contained in each factor, these have been denominated as follows:Digital competence perception during the pandemic (measures the valuation of the participants’ level of certain dimensions of their digital competence, such as student orientation, strategic vision, resilience, agility, or networking, during the pandemic).Professional aspects related to the use of DLE during the pandemic (measures the participants’ perception of the support received from students, fellow teachers and the university and their degree of satisfaction with the spaces and equipment available to them to face the digital challenges arising from the pandemic).Self-confidence during the pandemic (measures the participants’ degree of security, optimism and their feeling of control of the situation and the difficulties arising from the pandemic).Skills of adaptation to DLE (measures the degree of comfort and adaptability that participants express towards the DLE they have had to use during the pandemic, the degree of satisfaction with the training received in this regard, and the level to which the pandemic has led them to consider future goals for increasing their digital skills).Pandemic stress level in terms of professional work as professors (measures the impact on stress levels that the pandemic has had on the participants in their condition as university professors in terms of their feeling of tension, feeling upset, nervous, anxious or depressed, unable to cope with the consequences of the pandemic or to control the changes it causes and worried about contagion).

The first two factors are included within the digital competence variable, Factors 3 and 4 are integrated into the soft skills variable, while the last factor constitutes in itself the variable of psychological aspects related to the pandemic. The factors defined above explain 61.5% of the total variance (Table 4). The indices of the CFA confirm the theoretical model provided by the AFE, as the incremental fit indices are appropriate (AGFI = 0.8592; NFI = 0.7952; TLI = 0.9354; CFI = 0.8545; IFI = 0.8719) and the absolute fit indices are acceptable (GFI = 0.8504; RMSEA = 0.0356; AIC = 879.1549; chi-square/df = 3.1345).

Regarding the answers to the questions, it is necessary to observe that within Factor 5 (stress level) the high answers mean a high negative impact of the pandemic on the different studied affectivity dimensions of the participants. On the contrary, within the rest of the factors high answers mean high self-concepts of the participants regarding the aspects corresponding to each of them (level of digital competence, self-confidence, professional aspects of digital competence linked to the pandemic, and ability to adapt to DLE during the pandemic, respectively).

The five factors described above define what will henceforth be called subscales within the overall scale defined by the survey. The internal consistency of these subscales is high, given that all Cronbach’s alpha parameters and CR are above 0.70 (Table 5). The convergent validity analysis has been carried out through the AVE values (Table 5), which are adequate (exceeding 0.50 in all subscales).

The psychometric validation of the research instrument was carried out through Pearson correlation coefficients of the different subscales (Table 6). These statistics reveal that while the different subscales of the survey correlate weakly or very weakly with each other, the correlations of each subscale with the overall scale are high. All correlations are statistically significant at the 0.05 level of significance.

### 3.3. Analysis of the Answers to the Survey

Participants have an intermediate-high self-concept; at a general level, they show digital competencies and demonstrate ability to adapt to DLE during the pandemic within the soft skills variable (Table 7). Their average perception of self-confidence and professional aspects related to their digital competence during the pandemic is somewhat lower, and their valuation of the stress suffered is intermediate. The standard deviations are high (above a quarter of the mean), expressing a high dispersion in the answers (Table 7).

The degree of influence of perceived digital competence and self-confidence during the pandemic on the skills of adaptation to DLE and the influence of pandemic stress on all the factors above were studied through Pearson’s correlation coefficients (Table 6). In this sense, perceptions of one’s own digital competence and professional issues are related to the use of DLE influence professors’ assessment of their adaptation to the use of these environments during the pandemic. This influence is positive and moderate. The correlation between levels of self-confidence and adaptation to DLE is positive, though low. Professors’ pandemic stress has a negative influence on all the above factors (perception of digital competence, professional aspects, self-confidence and digital adaptability), though in a very weak way.

By digital generation, digital natives have a significantly higher self-concept in terms of their digital competence and their ability to adapt to DLE during the pandemic, although they report having suffered greater pandemic stress, as shown by the statistics of the *t*-test (Table 8). These mean differences occur in a situation of absence of homoscedasticity, except for the subscale of adaptation skills, for which homoscedasticity can be assumed, as shown by the Levene’s test statistics (Table 9). In fact, the answers of digital native professors are more heterogeneous than those of digital immigrant ones in all the subscales, except in that of adaptation skills, for which no significant differences have been identified for the standard deviations.

Pearson’s correlation coefficients between the different subscales of the survey (Table 10 and Table 11) show that the degree of dependence of the self-concept on their self-confidence, skills of adaptation to DLE during the pandemic, and digital competencies depends negatively and very weakly on pandemic stress among digital native professors. However, this degree of dependence is notably higher among digital immigrant professors in terms of dependence on digital competence, degree of self-confidence, and adaptation skills with respect to pandemic stress. Specifically, the negative correlation of pandemic stress and adaptability to DLE is five times higher in digital natives than in digital immigrants. Likewise, there is a positive and moderate influence of perceptions of digital competence and professional aspects in their assessment of adaptation to DLE during the pandemic. This correlation is higher in digital immigrants than in digital natives. Self-confidence positively influences digital adaptability, though weakly, and even more in digital natives than in digital immigrants.

Crossing the digital generation with the gender variable, the MANOVA test only identifies statistically significant differences in the subscales of professional aspects of the digital competence during the pandemic and stress level of the professors (Table 12). In both subscales, females offer higher self-concepts than males. Therefore, there is a gender gap in the perception of the professional aspects of digital competence in favor of females both within digital natives and digital immigrants. Among males digital natives express better self-concept in this subscale, while among females it is digital immigrants who manifest better self-concept in this regard. Moreover, the pandemic stress has been greater on females, on average, in both digital generations.

The area of knowledge of the participants is a characteristic of the academic profile of the participants that is highly discriminative of the dependent variables analyzed (Table 13). Digital natives express having better average digital competencies and professional aspect perceptions in all areas except in the area of Sciences and (only in terms of professional issues) in the area of Humanities. The area of Sciences is the only one in which digital natives express worse perceptions of their skills of adaptation to DLE than digital immigrants. In the areas of Social Sciences and Engineering, digital natives report lower self-confidence but better valuations of these adaptive skills than digital immigrants. Finally, in all areas of knowledge the pandemic stress on professors has been greater, according to the perception of the participants, in digital natives.

## 4. Discussion

Throughout this research the opinion of a group of digital immigrants and digital native university professors of their self-perception of their level of digital competence, self-confidence, skills of adaptation to DLE during the pandemic, professional aspects linked to the use of these environments, and stress caused by the COVID-19 pandemic has been collected. These data have been used to establish the degree of influence of university professors’ stress during the pandemic on their self-perception of digital competence, professional aspects, and self-confidence, and of the above factors on their perception of adaptation to digital learning environments. In addition, differences have been identified between the views expressed by professors of both digital generations in this regard. Consequently, the objectives set for the research have been achieved.

The digital competence of university professors has been shown to be key to generating sound learning and adequate academic performance in students [75,76]. Therefore, it is interesting to use instruments that assess the self-perception of this digital competence, such as, for example, the one used in this study, in order to analyze the influence of pandemic stress on the self-concept of professors’ digital competence [77].

Participants have shown, in general, moderate levels of stress due to the pandemic and somewhat higher levels of digital competence and adaptive skills to digital environments (Table 7). These levels of pandemic stress are consistent with results reported in other studies [14,15,36,78,79], although the populations studied were composed of teachers from different educational levels and geographic locations than in this study. This suggests the idea that the pandemic has indeed had a considerable psychological impact on education professionals worldwide. In [79] it is shown that there is a negative correlation between perceived self-efficacy in the use of ICT in virtual learning environments and pandemic stress, in a study of a population consisting of primary and secondary school teachers. Similarly, in [33] it is shown that levels of teaching digital competence are negatively correlated with pandemic stress in the context of Italian primary school teachers. When the population considered is composed of university students, negative correlations between pandemic stress and digital competence are obtained [80,81,82,83,84,85]. A similar trend is observed in the results of the present study, where the population is composed of university professors, as it was found that there is a negative correlation between pandemic stress and the rest of the analyzed variables (digital competence, professional aspects linked to it, self-confidence, and adaptation to digital environments in the pandemic). In the aforementioned variables, the correlation with pandemic stress is low in absolute value (Table 6). This result is in line with other studies framed in a population of university professors [86], although in those studies higher negative correlations are obtained.

It has been shown that pandemic stress has a greater negative impact on professors’ self-confidence as well as that the self-perception of skills of adaptation to digital environments has a positive, although moderate, correlation with the self-concept of digital competence. It can be concluded that, in general, professors’ pandemic stress negatively influences their self-confidence as well as their perception of digital competence and their adaptation to digital environments due to the pandemic, although these latter two only very weakly,. The weakness of the above correlations together with the fact that the ratings given by the professors to their digital adaptation are intermediate (Table 7) suggests the idea that there may be sociological factors (such as differences in the confinement measures carried out in different countries, the way in which vaccination has developed, etc.) or academic factors (such as the training and digital adaptation measures carried out by the different universities) that may be influencing the digital adaptation of the professors. Thus, the identification of these factors constitutes an interesting line of future research.

As far as it has been possible to explore, there are no previous studies on the impact of the pandemic on the digital competence of university professors differentiated by the digital generation to which they belong. However, there are studies that analyze the influence of being a digital native on the development of digital competence and its adaptability in the management of virtual learning environments in students after or during the pandemic [87,88]. Likewise, in [89,90] it is shown that within a population of future teachers those who are digital natives show better skills in terms of DLE use and worse social skills in online interaction in discussion forums. Other studies argue that digital immigrant status hinders the acquisition of adequate digital skills for full incorporation of the use of DLE [91]. The results of these studies in the student population coincide with those presented here for university professors (Table 8 and Table 9), in the sense that the perception of digital competence is better in digital natives. Furthermore, the literature supports a positive correlation between the digital competencies of university professors and those of their students; the greater digital competency of the former results in greater digital competency in the latter [75].

This work corroborates the fact, ascertained in the preceding literature, that digital native university professors show greater digital competence than professors who are digital immigrants. In addition, it is shown that there is a positive and moderate correlation between professors’ self-confidence and self-concept about their ability to adapt to digital environments and their perception of their own digital competence, a correlation that is higher in digital immigrants than in digital natives (Table 10 and Table 11). Despite this, pandemic stress is higher in digital natives, who show lower self-confidence than digital immigrants (Table 8). This is probably due to the older age and consequently greater professional experience of digital immigrants. The influence of pandemic stress is greater in digital natives in terms of both their perception of their digital adaptation skills and in their perception of their digital competence. On the other hand, there is a strong gap by area of knowledge (Table 13) and a weak gap by gender (Table 12), the origins of which require further study. All these results are novel in the specialized literature and demonstrate the original contribution of the present work.

From all of the above, it follows that it is advisable for universities to design continuous training programs for university professors focused on increasing their ability to adapt to DLE. This suggestion is in line with other studies that have shown that digital competence training for university professors is effective in increasing their digital skills [92,93,94,95,96]. Even in Health Sciences, where there has been some reluctance to use digital learning environments, it has been shown that the incorporation of these technologies has a formative benefit for students [91]. This training should be more focused on reinforcing digital competence in the case of digital immigrant professors, given that they report lower levels of digital competence and self-confidence than digital natives. It should be noted that, on the other hand, pandemic stress affects them to a lesser extent, though this impact negatively influences their digital competence more than in digital natives. In the case of digital natives, however, reinforcing levels of self-confidence could be more effective in balancing their competencies in adapting to digital learning environments.

## 5. Conclusions

Throughout this research it has been shown that digital native university professors have a significantly higher self-concept about their level of digital competence and their ability to adapt to virtual learning environments than digital immigrant professors, although the latter report higher self-confidence than digital natives. The stress that the pandemic originated by COVID-19 has caused is significantly higher in digital natives as well. In addition, digital immigrant professors express a greater influence of self-confidence and coping skills on their level of digital competence than digital natives. Consequently, there is a moderate significant influence between the self-concept of soft skills and that of digital competence. This influence is more clearly perceived by digital immigrants, among whom, moreover, the levels of digital competence and soft skills are more influenced by the psychological impact of the pandemic.

In the two digital generations analyzed, the variable measuring the professors’ area of knowledge (a variable of an academic nature) is more discriminative than other variables of a sociological nature, such as gender. It is worth noting that in all areas of knowledge except for Science, digital natives express a better self-concept of both their soft skills and their digital competence. The exception to this conclusion is that in Social Sciences and Engineering digital natives express lower self-confidence than digital immigrants.

## 6. Limitations, Further Research, and Recommendations

The main limitations of this study involve the methodology used, which is quantitative, and the geographic area of the population studied. Consequently, further research could introduce qualitative methodologies to explore the reasons underlying the influence of the variables analyzed here on the perceptions of digital teaching stress in university professors and their adaptability to digital learning environments. In this sense, a more in-depth study would be necessary to identify the peculiarities that make the area of Science behave in the opposite way to the rest of the knowledge areas with respect to the self-concept of soft skills and digital competence. In addition, it would be interesting to extend the sample to wider geographical areas and thus analyze differences in terms of adaptability to digital environments according to different geographical zones.

It is recommended that university professors participate in training courses on digital training specifically oriented to their teaching activity. These training actions should be designed taking into consideration the characteristics of each area of knowledge and the specific needs of different age ranges. Finally, from the research point of view, it is necessary to develop strategies to prevent the emergence of digital stress phenomena related to teaching in digital learning environments.

## Figures and Tables

**Figure 1 ijerph-19-03732-f001:**
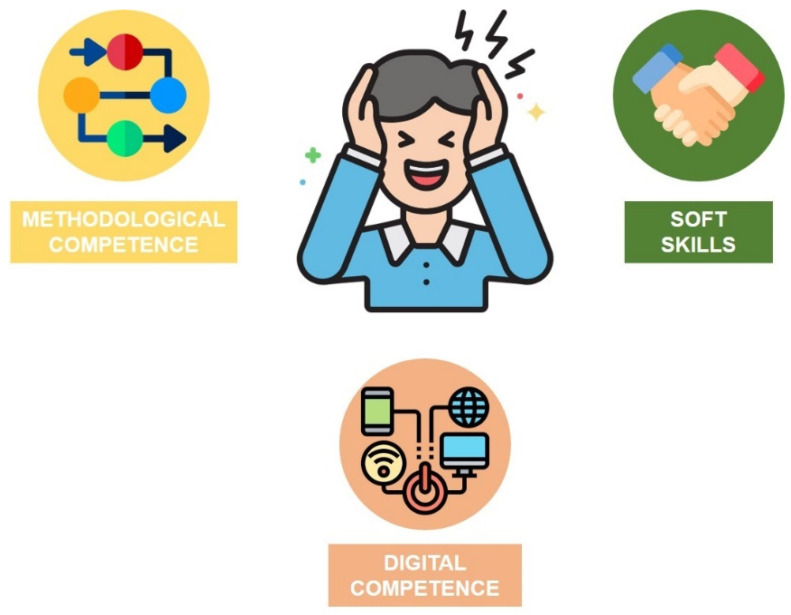
Teacher’s skills and competences to cope with COVID-19 pandemic and the virtualization of education.

**Figure 2 ijerph-19-03732-f002:**
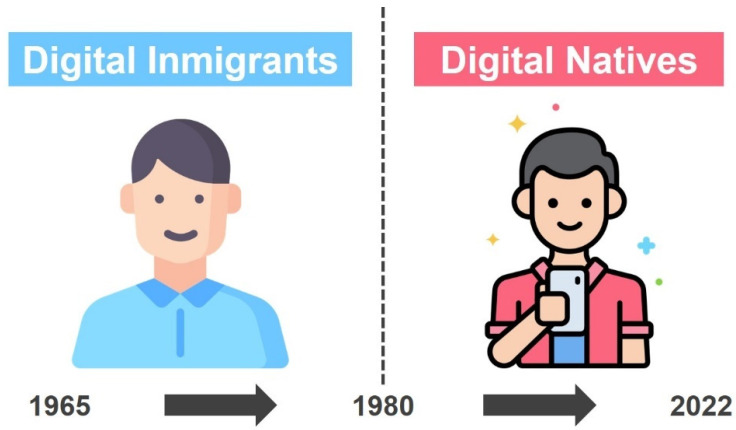
Digital generations.

**Figure 3 ijerph-19-03732-f003:**
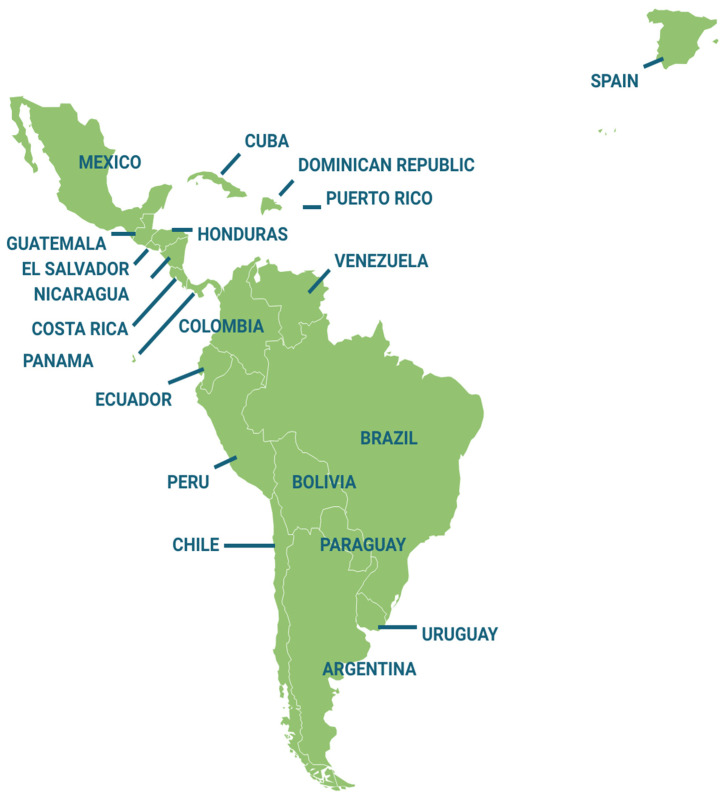
Countries participating in the research.

**Figure 4 ijerph-19-03732-f004:**
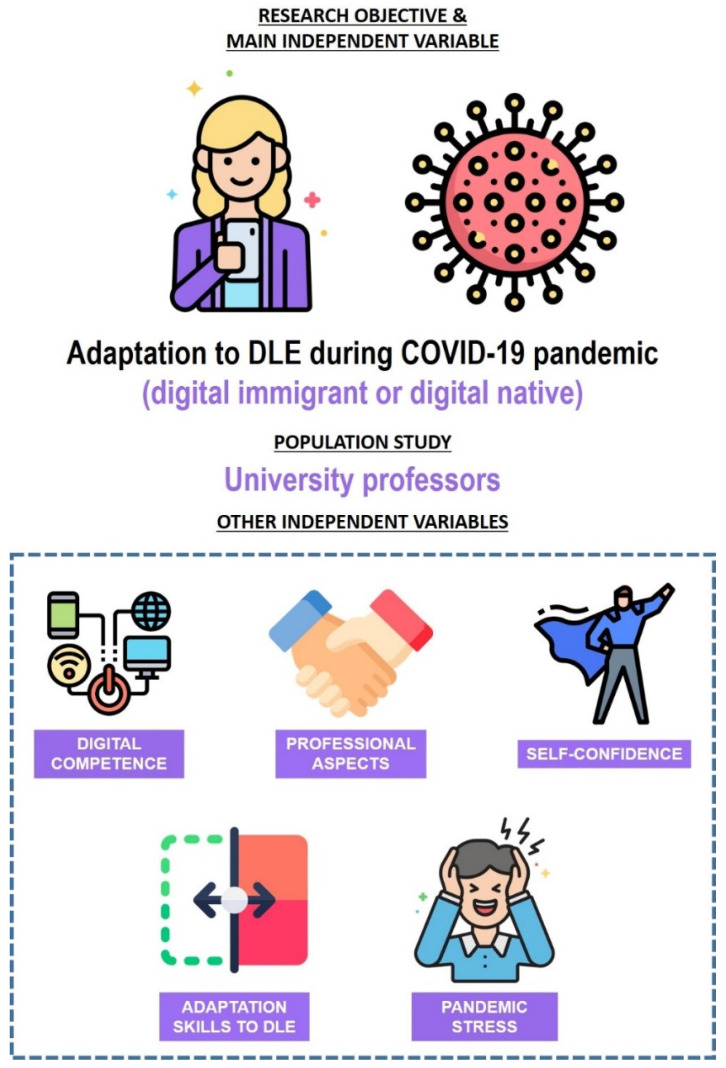
Research objective and variables.

**Table 1 ijerph-19-03732-t001:** Items of the survey.

Variable	Item	Question
Assessment of the aspects indicated on digital competence	Item 1	Agility
	Item 2	Continuous learning
	Item 3	Digital communication
	Item 4	Information management
	Item 5	Network leadership
	Item 6	Student orientation
	Item 7	Resilience
	Item 8	Teamwork
	Item 9	Strategic vision
Assessment of the aspects indicated on professional aspects of the use of DLE	Item 10	Student support
	Item 11	Faculty support
	Item 12	University support
	Item 13	Spaces used
	Item 14	Technical equipment used
Self-confidence during the pandemic	Item 15	I have been confident about my ability to handle my personal problems related to the pandemic
	Item 16	I have felt optimistic during the pandemic
	Item 17	I have felt that I can control the difficulties that could appear in my life due to the infection
	Item 18	I have felt that I have everything under control in relation to the pandemic
Adaptation skills to DLE during the pandemic	Item 19	I have fun and feel comfortable when teaching through DLE during the pandemic
	Item 20	Adaptation as a teacher to the use of DLE during the pandemic
	Item 21	Training received on digital competence during the pandemic
	Item 22	I consider developing objectives and activities in the future to increase my digital competence in the wake of the pandemic
Assessment of the indicated aspects of pandemic stress in the work of professors	Item 23	I feel tense
	Item 24	I feel that difficulties are piling up
	Item 25	I am upset because I am not in control of the pandemic aspects of the pandemic
	Item 26	I feel affected, as if something serious is going to happen to me unexpectedly
	Item 27	I feel unable to cope with the circumstances of the pandemic
	Item 28	I feel stressed by the pandemic
	Item 29	I feel unable to control important aspects of life because of the pandemic
	Item 30	I think I suffer from anxiety
	Item 31	I feel depressed
	Item 32	I am afraid of infection

**Table 2 ijerph-19-03732-t002:** Distribution of the participants by gender and area of knowledge, differentiating by digital generation (in percentages) and statistics of the Pearson independence test.

		Immigrants (%)	Natives (%)	Pearson’s Chi-Square	*p*-Value
Gender	Males	46.1	37.5	75.644	0.0000 *
	Females	53.9	62.5		
Area of knowledge	Arts and Humanities	13.9	26.8	424.13	0.0000 *
	Sciences	11.3	16.1		
	Health Sciences	13.0	10.7		
	Social and Legal Sciences	33.9	20.5		
	Engineering and Architecture	27.8	25.9		

* *p* < 0.05.

**Table 3 ijerph-19-03732-t003:** Exploratory Factor Analysis of the survey.

Item	Factor 1 Digital Competence	Factor 2 Professional Aspects	Factor 3 Self-Confidence	Factor 4 Adaptation Skills	Factor 5 Stress Level
Item 1	0.788				
Item 2	0.798				
Item 3	0.752				
Item 4	0.838				
Item 5	0.758				
Item 6	0.732				
Item 7	0.735				
Item 8	0.733				
Item 9	0.794				
Item 10		0.604			
Item 11		0.672			
Item 12		0.702			
Item 13		0.535			
Item 14		0.542			
Item 15			0.652		
Item 16			0.726		
Item 17			0.676		
Item 18			0.661		
Item 19				0.770	
Item 20				0.713	
Item 21				0.603	
Item 22				0.653	
Item 23					0.519
Item 24					0.677
Item 25					0.646
Item 26					0.729
Item 27					0.580
Item 28					0.833
Item 29					0.668
Item 30					0.692
Item 31					0.585
Item 32					0.559

**Table 4 ijerph-19-03732-t004:** Cumulative proportion of explained variance of the principal component analysis.

Item	Factor 1 Digital Competence	Factor 2 Professional Aspects	Factor 3 Self-Confidence	Factor 4 Adaptation Skills	Factor 5 Stress Level
Proportion Variance	0.296	0.067	0.078	0.062	0.112
Cumulative Variance	0.296	0.363	0.441	0.503	0.615

**Table 5 ijerph-19-03732-t005:** Cronbach’s alpha parameters of the subscales.

Subscale	Cronbach’s Alpha	CR	AVE
Technical aspects	0.90	0.88	0.67
Professional aspects	0.82	0.80	0.60
Self-confidence	0.82	0.82	0.61
Adaptation skills	0.74	0.72	0.52
Psychological impact	0.88	0.87	0.66

**Table 6 ijerph-19-03732-t006:** Pearson correlation of the subscales among themselves and with respect to the global scale.

	Digital	Professional	Self-Confidence	Adaptation	Stress	Global
Digital	1	0.3021	0.2672	0.3786	−0.0688	0.8219
Professional		1	0.1713	0.3054	−0.0095	0.7622
Self-confidence			1	0.1203	−0.1169	0.7011
Adaptation				1	−0.0617	0.7740
Stress					1	−0.7097
Global						1

**Table 7 ijerph-19-03732-t007:** Mean values and standard deviations of the subscales of the survey (ratings out of 5).

	Mean Values	Standard Deviations
Digital competence	3.79	0.94
Professional aspects	3.44	1.16
Self-confidence	3.16	1.06
Adaptation skills	3.87	0.96
Stress level	2.54	1.18

**Table 8 ijerph-19-03732-t008:** Mean values of the subscales of the survey and statistics of the *t*-test when differentiating by digital generation (mean values out of five).

	Digital Immigrants	Digital Natives	*t*	*p*-Value
Digital competence	3.66	3.92	−14.101	0.0000 *
Professional aspects	3.42	3.46	−1.2881	0.1978
Self-confidence	3.21	3.11	2.7746	0.0056 *
Adaptation skills	3.83	3.91	−2.2340	0.0256
Stress level	2.38	2.70	−12.940	0.0000 *

* *p* < 0.05.

**Table 9 ijerph-19-03732-t009:** Standard deviations of the subscales of the survey and statistics of the Levene’s test when differentiating by digital generation (deviations out of five).

	Digital Immigrants	Digital Natives	Levene’s F	*p*-Value
Digital competence	0.99	0.88	71.559	0.0000 *
Professional aspects	1.20	1.11	7.0268	0.0081 *
Self-confidence	1.10	1.01	17.924	0.0000 *
Adaptation skills	0.97	0.96	1.3159	0.2514
Stress level	1.14	1.19	42.225	0.0000 *

* *p* < 0.05.

**Table 10 ijerph-19-03732-t010:** Pearson’s correlation coefficients of the different subscales of the survey within digital immigrant professors.

	Digital	Professional	Self-Confidence	Adaptation	Stress
Digital	1	0.3001	0.3048	0.3847	−0.1043
Professional		1	0.1727	0.3786	−0.0024
Self-confidence			1	0.1063	−0.1335
Adaptation				1	−0.1072
Stress					1

**Table 11 ijerph-19-03732-t011:** Pearson’s correlation coefficients of the different subscales of the survey within digital native professors.

	Digital	Professional	Self-Confidence	Adaptation	Stress
Digital	1	0.3688	0.2347	0.3688	−0.0573
Professional		1	0.1404	0.2208	−0.0333
Self-confidence			1	0.1404	−0.0881
Adaptation				1	−0.0264
Stress					1

**Table 12 ijerph-19-03732-t012:** Mean values of the subscales of the survey differentiating by digital generation and gender and statistics of the MANOVA test (mean values out of five).

	Males		Females		MANOVA	*p*-Value
	Digital Immigrants	Digital Natives	Digital Immigrants	Digital Natives		
Digital competence	3.66	3.88	3.66	3.95	3.0027	0.0832
Professional aspects	3.31	3.45	3.51	3.45	6.0071	0.0143 *
Self-confidence	3.29	3.17	3.14	3.08	0.8803	0.3482
Adaptation skills	3.80	3.83	3.87	3.95	0.5190	0.4713
Stress level	2.50	2.53	2.27	2.79	98.275	0.0000 *

* *p* < 0.05.

**Table 13 ijerph-19-03732-t013:** Mean values of the subscales of the survey differentiating by digital generation and area of knowledge (I. means digital immigrants and N. means digital natives) and statistics of the MANOVA test (mean values out of five).

	Humanities		Sciences		Health		Social Sci.		Engineering		MANOVA	*p*-Value
	I.	N.	I.	N.	I.	N.	I.	N.	I.	N.		
Digital	3.76	4.02	3.73	3.66	3.26	3.94	3.70	4.11	3.72	3.84	33.400	0.0000 *
Professional	3.59	3.58	3.35	3.17	3.29	3.68	3.42	3.49	3.41	3.41	4.8219	0.0007 *
Self-confidence	3.06	3.10	3.10	2.75	2.90	3.35	3.30	3.26	3.37	3.14	10.763	0.0000 *
Adaptation	3.88	3.92	4.12	3.56	3.60	3.92	3.72	3.93	3.95	4.09	16.6491	0.0000 *
Stress	2.28	2.77	2.28	2.54	2.35	2.63	2.35	2.57	2.52	2.84	3.6002	0.0061 *

* *p* < 0.05.

## Data Availability

The data are not publicly available because they are part of a larger project involving more researchers. If you have any questions, please ask the contact author.

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
