# Peer review of "Variables Influencing Professors’ Adaptation to Digital Learning Environments during the COVID-19 Pandemic"

_ijerph, 2022, doi:10.3390/ijerph19063732_

Round 1
Reviewer 1 Report
Regarding the content, I do not have any changes to recommend, it makes a good literary review to support the relevance of the problem to be studied and a good structuring of the content, it uses the correct methodology for this type of study and it is a consistent and well-detailed methodology to give significance to the results they show, makes a good discussion of the results with respect to the studies carried out previously, and marks the conclusion obtained well.
Although I advise looking at these things:
Never two sections without a paragraph of text in between. You should put a couple of lines describing/naming the subsections you are going to deal with within that section. You must correct this between lines 147-148 and 227-228.
And the section “5. Conclusions” should also include the implications of this study, the limitations and future lines of research opened with this study. This section should be 1 page maximum.
Author Response
Please, find attached the reponse.

Reviewer 2 Report
Congratulations to the authors. It is a good manuscript. Quite current, but there are already many publications related to this topic. Still, it is an original manuscript.
The aim of this manuscript was to find out how research on teacher stress and anxiety associated with the use of educational technology was proceeding.
The main findings also show that teachers present high levels of anxiety or stress due to their use of educational technology in the classroom. Among the conclusions, the need for research on different strategies to prevent the emergence of these anxiety and stress symptoms in teachers stands out.
We believe that adding this reference enhances and further contextualizes your manuscript.
Congratulations to the editors and authors.
Author Response
Please, find attached the response.

Reviewer 3 Report
a) In defining digital immigrants, please bear in mind that although age is an important factor, it is not the only major factor. The other important factor is socio-cultural context of participants. Students from low income contexts may be digital immigrants by reason of having had no exposure to ICT despite their young age
b) Would be a good idea to have a section for recommendations towards the end. For instance, what could be the focus of further research? Secondly, what do the findings of the study suggest should be done by practitioners in view of what the study brought out?
This is an excellent manuscript. It is readable, logical, and relevant. The inclusion of images is a great idea.
Author Response
Please, find attached the response.
